# Application of Improved Quasi-Affine Transformation Evolutionary Algorithm in Power System Stabilizer Optimization

**Jing Huang, Jiajing Liu, Cheng Zhang, Yu Kuang and Shaowei Weng ***

School of Electronic, Electrical Engineering and Physics, Fujian University of Technology, Fuzhou 350118, China
* Correspondence: wswweiwei@126.com

**Abstract:** This paper proposes a parameter coordination optimization design of a power system stabilizer (PSS) based on an improved quasi-affine transformation evolutionary (QUATRE) algorithm to suppress low-frequency oscillation and improve the dynamic stability of power systems. To begin, the simulated annealing (SA) algorithm randomly updates the globally optimal solution of each QUATRE iteration and matches the inferior solution with a certain probability to escape the local extreme point. This new algorithm is first applied to the power system. Since the damping ratio is one of the criteria with which to measure the dynamic stability of the power system, this paper sets the objective function according to the principle of maximization of the damping coefficient of the electromechanical mode, and uses SA-QUATRE to search a group of global optimal PSS parameter combinations to improve the safety factor of the system as much as possible. Finally, the method's rationality and validity were validated by applying it to the simulation examples of the IEEE 4-machine 2-area system with different operation states. The comparison with the traditional optimization algorithm shows that the proposed method has more advantages for multi-machine PSS parameter coordination optimization, can restrain the low-frequency oscillation of the power system more effectively and can enhance the system's stability.

**Keywords:** simulated annealing quasi-affine transformation evolutionary (SA-QUATRE); coordinated optimization design; power system stabilizer

## 1. Introduction

In recent years, the continuous operation of large generator sets and the large-scale use of high-gain and fast excitation systems have improved the transient level of the system, but also caused the problem of low frequency oscillation of the power system due to insufficient damping, which has increasingly become an important factor restricting the overall interconnection of the power grid [1–3]. A power system stabilizer (PSS) can not only offset the negative damping torque generated by the regulator, but also provide additional positive damping for the system, which is the most efficient and economical measure for suppressing the low-frequency oscillation of the power system [4,5]. Especially in the context of a large power grid, PSS has been widely used. The configuration of PSS, that is, the research theory of the installation site, has been perfected, and another key issue is the parameter coordination, which is also one of the hot topics discussed by scholars. It has been shown that the damping of local and interval oscillation modes can be optimized to a certain extent if the PSS can obtain appropriate parameters [6].

In control theory, the phase compensation approach can be used to design a power system stabilizer, but for multiple power system stabilizers in the system, the traditional phase compensation method usually carries out parameter setting sequentially, which will inevitably cause the interaction effect between each stabilizer [7]. To overcome this phenomenon, some studies have pointed out that the parameter tuning of PSS is equivalent to an optimal problem based on the eigenvalues of electromechanical modes, that is, to move

the eigenvalues to the left half plane of the complex plane as much as possible. In recent years, using the global optimization ability of artificial intelligence algorithms to solve this phenomenon has provided a new idea for the coordination design of a controller and has become a hot research topic. Particle swarm optimization (PSO), a basic and powerful approach that many researchers have studied and applied to power systems, is a commonly used optimization algorithm [8]. In Ref. [9], PSS parameters were optimized based on a PSO algorithm. Although the optimization principle is simple, the optimized particles in the optimization iteration process are prone to precocity and easily fall into local optimum, which is also the common fault of most algorithms. For this reason, Ref. [10] proposed an improved PSO algorithm based on particle swarm theory. Although the performance of the algorithm has been improved to a certain extent, the particles do not completely tend to the global optimal value, but tend to a relatively good value. Ref. [11] proposed the gray wolf optimization algorithm to optimize PSS parameters, which has fewer algorithm parameters and faster convergence speed. However, in the later optimization period, with the decrease of population number, the optimization rate decreases, and at the same time, it is possible to fall into local optimal. Therefore, Ref. [12] proposed an improved gray wolf optimization algorithm, which overcomes the problem that it is easy to fall into local optimum and strengthens the robustness of the algorithm. However, the optimization algorithm search process tends to be complicated, so the reliability of the optimization result needs to be further improved.

In this paper, a method combining the quasi-affine transformation evolutionary algorithm [13] and the simulated annealing algorithm is proposed to coordinate the optimization design of stabilizer parameters of multi-machine power system. The main contribution of the paper lies in the following:

The proposed SA-QUATRE algorithm not only overcomes the defect of slow convergence of particle cluster optimization algorithm, it also has stronger cooperation and reduces time complexity, and accepts inferior solutions with a certain probability to avoid falling into local minimum in the process of searching.

To overcome the interaction between power system stabilizers, in this paper, the parameter coordination tuning problem of the power system stabilizer is transformed into the optimal problem based on the characteristic value of oscillation mode and damping coefficient, and the dynamic stability of the power system is improved.

The validity and feasibility of the proposed method are verified by simulation with a test function and the IEEE 4-machine 11-node system. It can be found that, compared with the existing multi-machine power system with PSS parameters, the multi-machine and multi-node power systems with the SA-QUATRE algorithm designed and optimized under stabilizer suppression has better system dynamic performance.

The rest of the paper is organized as follows: Section 2 introduces PSS structure and each link, as well as its transfer function, objective function and constraints of the parameter optimization of PSS for the multi-machine system. Section 3 introduces the proposed SA-QUATRE algorithm in detail and PSS parameter optimization based on it. Section 4 discusses the simulation example and the experiment results. Section 5 presents the final conclusion.

## 2. Parameter Optimization of PSS for Multi-Machine System

Voltage stabilizers suppress oscillations by adding a signal to the excitation system that produces a positive damping torque to offset the negative damping torque produced by the Voltage regulator [14]. The principle and excitation systems are shown in Figure 1, where, taking the rotor angle deviation $\Delta w$ of the generator as the input signal, the PSS and excitation system transfer function can be expressed as [15]:

$$U_{PSS,i} = K_{s,i} \frac{sT_{5,i}}{1 + sT_{5,i}} \left( \frac{1 + sT_{1,i}}{1 + sT_{2,i}} \right) \left( \frac{1 + sT_{3,i}}{1 + sT_{4,i}} \right) \cdot \Delta w, \tag{1}$$

$$E_{fd,i} = \frac{K_{A,i}}{1 + sT_{A,i}}(U_{REF,i} - U_C + U_{PSS,i}), \tag{2}$$

where the subscript $i$ represents the generator $i$; $K_s$ is the amplification gain, $T_5$ is the high-pass filter time constant, usually set as 5 s or 10 s; in this paper, $T_5 = 10\ s$. Two lead-lag compensators are used to eliminate the delay between excitation and electromagnetic torque. In practical applications, the two lead-lag compensators can compensate the low frequency and high frequency phases respectively. After each input signal passes through PSS, its output signal can provide the corresponding reference voltage for the excitation system, and its reference voltage serves as the reference modulation signal provided to the excitation system, so that the negative damping or weak damping caused by the fast excitation system can be compensated accordingly [16,17].

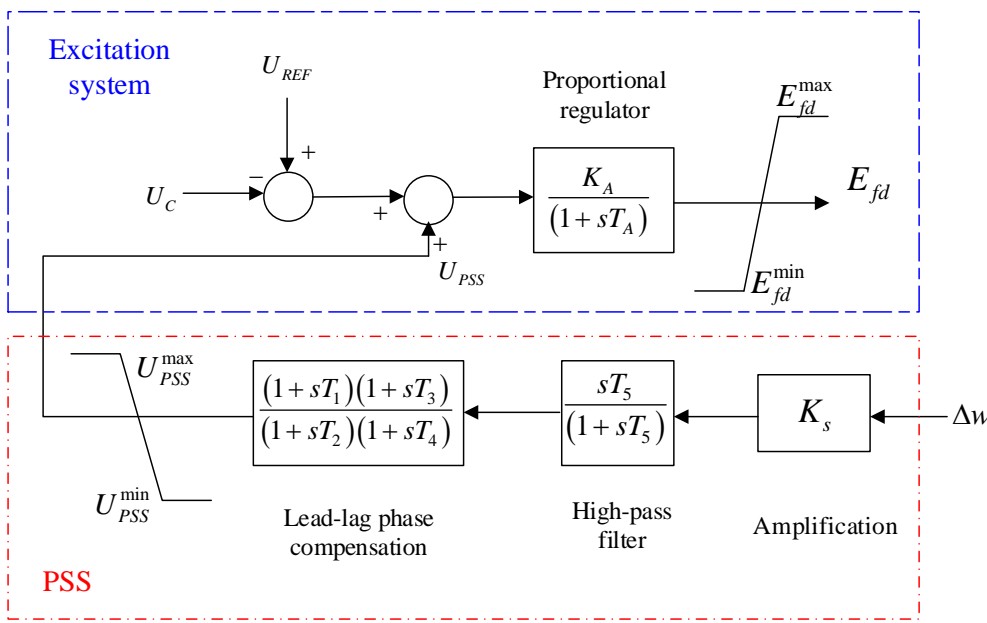

**Figure 1.** Excitation system with traditional lead-lag compensation type PSS.

PSS plays an effective role mainly by adjusting $K_s$ and the inertia time constant of the lead-lag compensators, and the other parameters for a generator are usually unchanged. Therefore, it is necessary to tune parameters reasonably, otherwise the effect of suppressing low frequency oscillation will be counterproductive.

In general, a linear differential equation can be used to describe the dynamic behavior of a power system:

$$\dot{x} = f(x, u), \tag{3}$$

where the column vector $x$ is the state vector, $u$ is the external input vector, and the derivative of the state variable to time is represented by $\dot{x}$. When the variables of the system remain constant, the system is in equilibrium. The nonlinear system equation can then be linearized at the equilibrium point [18], where the equation is as follows:

$$\begin{cases} \Delta\dot{x} = A\Delta x + B\Delta u, \\ \Delta y = C\Delta x + D\Delta u, \end{cases} \tag{4}$$

where $\Delta x$ is the state vector, $\Delta y$ is the output vector, $\Delta u$ is the input vector, $A$ is the state matrix, $B$ is the control matrix, $C$ is the output matrix, $D$ is the feedforward matrix. For the power system, $y$ is not a direct function of $u$, that is $D = 0$. The stability of the system can be determined by judging the position of eigenvalues of the state matrix $A$ on the complex plane [19].

Many studies show that the generator controller is effective in changing the real part of the oscillation mode; in the meantime, it has little influence on the imaginary part.

As shown in Figure 2, the dotted line is equal to the damping ratio line. After a lot of engineering practice, the Ontario Electric Power Bureau of Canada proposed that $\xi = 0.03$ is the critical state in the normal operation of the power system. Thus, $\xi_{\min}$ is set as 0.03 in this paper [20]. In fact, the optimization of PSS is to move the eigenvalues of the state matrix of the system to the left part of the complex plane through continuous optimization, as far away from the virtual axis as possible. According to this rule, we can define the optimization objective function of the stabilizer and measure the dynamic characteristics of the system by the damping ratio of the system.

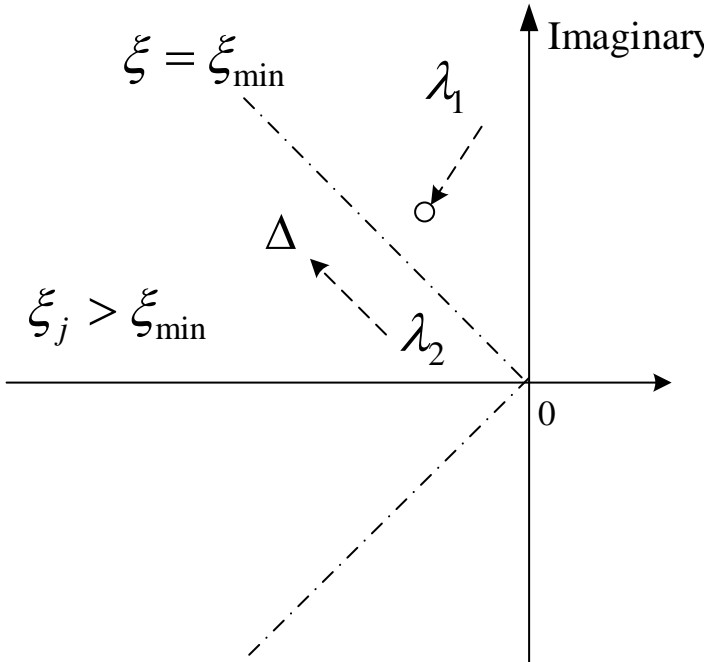

**Figure 2.** The left region of s-plane.

In the complex plane, each such conjugate complex root $\lambda_j = \sigma_j + jw_{dj}$ has the following relationship with the system damping ratio $\xi_j$:

$$\xi_j = \frac{-\sigma_j}{\sqrt{\sigma_j^2 + w_{dj}^2}}. \tag{5}$$

Define the objective function $f$:

$$f = \sum_{j=1}^{n} \sum_{i=1}^{k} \left( \xi_{\min} - \xi_{i,j} \right), \tag{6}$$

where $n$ represents the number of operating modes, $k$ is the number of oscillation modes, $\xi_{\min}$ is the preset minimum damping coefficient, $\xi_{i,j}$ represents the $i$th electromechanical oscillation mode damping coefficient in the $j$th operating mode. Combined with other constraints, the parameter coordination of PSS can be expressed as the following eigenvalue optimization form:

$$\begin{cases} \min f, \\ s.t.\ K_{i\min} \leq K_i \leq K_{i\max}, \\ T_{1i\min} \leq T_{1i} \leq T_{1i\max}, \\ T_{3i\min} \leq T_{3i} \leq T_{3i\max}, \end{cases} \tag{7}$$

where $K_{i\max} = 50$, $K_{i\min} = 0.1$, $T_{1i\max} = T_{3i\max} = 1$, $K_i$, $T_{1i}$ and $T_{3i}$ are three variables to be optimized and the remaining parameters are given in advance.

## 3. PSS Parameter Optimization Based on SA-QUATRE Algorithm

The evolution formula of the quasi-affine transformation is analogous to the affine transformation in geometry. The affine transformation function $(f : X \mapsto Y)$ in geometry is as follows:

$$X \mapsto MX + B. \tag{8}$$

The evolutionary structure of the QUATRE algorithm is to use $X \mapsto MX + \overline{M}B$. Supposing a population of N particles is searching in a D-dimensional space, and $X$ in Formula (8) is used to represent the particle's position. If the position of particle $i$ is $X_i = [x_1, x_2, \ldots, x_D]$, and the population size is ps, the population position can be expressed as $X = [X_1, X_2, \ldots, X_{ps}]^T$. Then the particle position update formula is as follows:

$$\begin{cases} X \mapsto M \otimes X + \overline{M} \otimes B, \\ B = X_{gbest} + c * (X_{r1} - X_{r2}), \end{cases} \tag{9}$$

where $B$ represents the evolutionary guidance matrix, $\otimes$ represents the bitwise multiplication of matrix elements. $X_{r1}$ and $X_{r2}$ are generated by randomly arranging the row vectors of the matrix $X$. Their difference is the difference matrix, which is used to represent the particle search radius. This search method helps to adapt to different search dimensions, $c$ is the coefficient factor or step size of the differential matrix. If the $i$th particle obtains the best fitness value, it is recorded as $X_{best_i}$. Then the global best coordinate matrix for each particle is shown below, and its size is $ps * 1$.

$$X_{best} = [X_{best_i}, X_{best_i}, \ldots, X_{best_i}]^T. \tag{10}$$

$M$ is the cooperative search matrix. $\overline{M}$ represents the incidence matrix of $M$ which is the core of the algorithm. This algorithm transforms all the ordinary individuals of the population into the optimal individuals and extends the global optimal individual search method to the whole population. Although this operation achieves individual equivalence, the complexity of the algorithm will become $n$ times. Therefore, the collaborative architecture is introduced to reduce the search complexity of the algorithm, which is implemented by the $M$ matrix.

The $M$ matrix is transformed from its initialization matrix $M_{init}$. $M_{init}$ is a D-dimensional Boolean matrix whose elements of the lower triangular matrix are all 1 and the rest elements are 0, as shown in Equation (11). Its stacking method is determined according to the population size ps, and the dimension $D$. There are three stacking methods: if ps = D, if ps = n * D, $M_{init}$ is n times the previous case of vertical stack, if ps = n * D+ k and ps%D = k (% means remainder), the $n * ps$ row of $M_{init}$ is consistent with the second method, and the last $k$ rows are the first $k$ rows of the Boolean matrix in the first method. After the stacking is completed, the conversion is achieved through two consecutive operations. The first step is to randomly arrange the row elements of the matrix $M_{init}$ and perform independent operations on each row element, The second step is to randomly arrange the row vectors of the matrix without changing the row elements [21]. So, $M$ is compared with $M_{init}$, the only thing that changes is the position of 0 and 1 in each row and the position of each row vector.

$$M_{init} = \begin{bmatrix} 1 & & & \\ 1 & 1 & & \\ & & \cdots & \\ 1 & 1 & \cdots & 1 \end{bmatrix} \sim \begin{bmatrix} & 1 & & \\ & \cdots & & \\ 1 & & \cdots & 1 \\ & 1 & 1 & \end{bmatrix} = M. \tag{11}$$

This cooperative structure can effectively solve the defects that individuals in the population cannot achieve information sharing and high coordination in the evolution process due to the existence of two states of global search and local search.

Although QUATRE can gain performance advantages and reduce time complexity by multiplying the population, it will also fall into a local optimum in the later stage of the algorithm due to the reduction of population diversity. The simulated annealing algorithm

is a global search algorithm, which accepts the difference with a certain probability during the search process and jumps out of the local optimum. Based on selecting the appropriate temperature parameter T, this paper uses a simulated annealing algorithm to improve QUATRE, which can improve the deficiencies of QUATRE's later local optimization, speed up the algorithm's process, and better design the parameters of the power system stabilizer. Figure 3 shows the flow of the improved quasi-affine transformation evolution algorithm. The steps are summarized as follows:

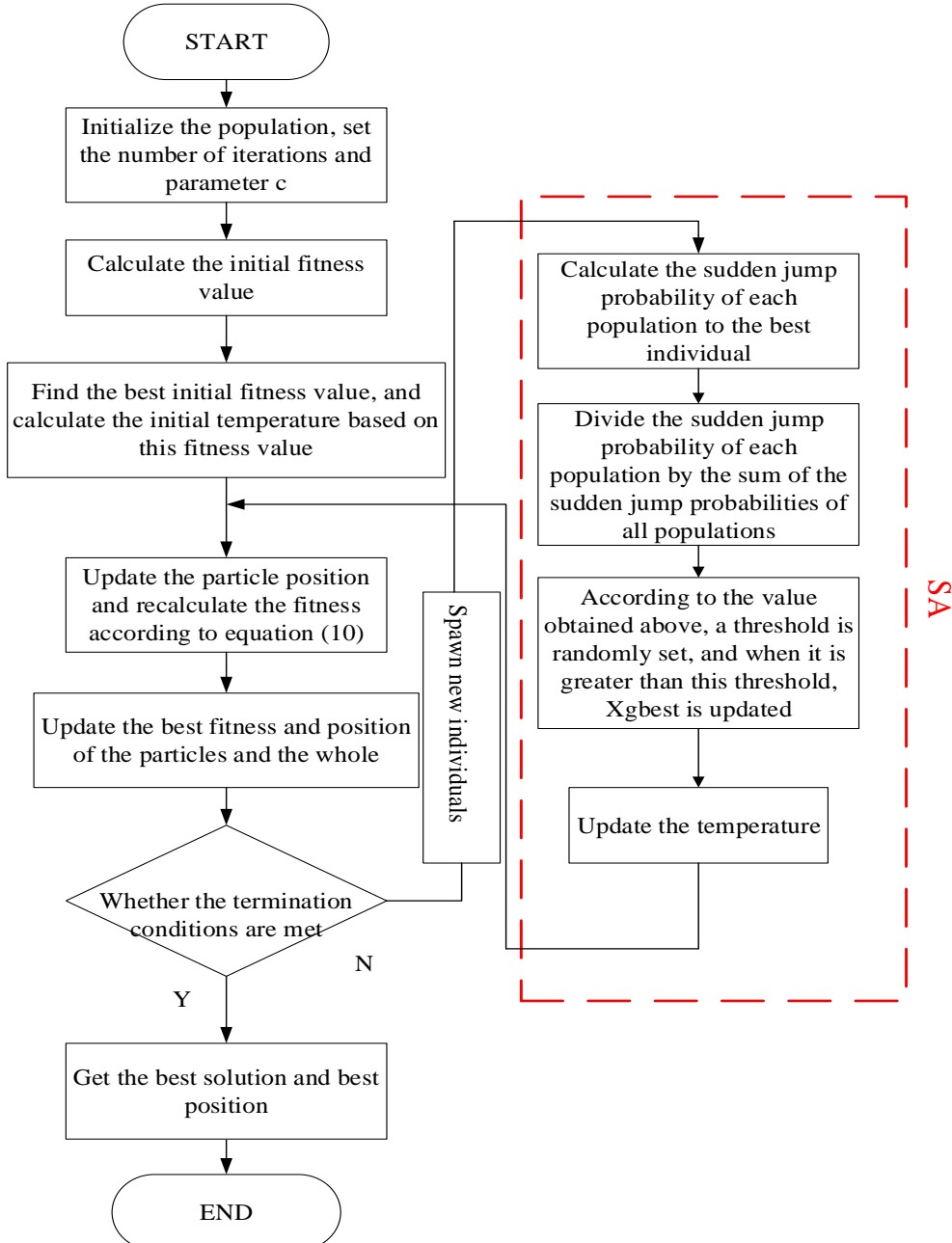

**Figure 3.** Flow chart of SA-QUATRE algorithm.

Step1: Initialize the population and randomly initialize the particle position $X = [X_1, X_2, \ldots, X_N]^T$, limit the search range, set the difference matrix coefficient factor $c$, and set the initial position as the individual historical optimal solution $X_{pbest}$ of each particle as and the optimal global solution $X_{gbest}$;

Step2: Calculate the objective function $fitness(X_i)$ of each particle;

Step3: Find the optimal value $fitness_{\min}(X)$ in the value of the initial particle objective function, and calculate the initial temperature $T$ of the simulated annealing, $T = -fitness_{\min}(X) / \log(0.2)$;

Step4: Calculate the sudden change probability at each temperature. The method is first to calculate the sudden change probability of each population corresponding to the best individual and then divide by the sum of the jump probability of all populations;

Step5: Randomly set and generate a probability. If the jump probability of a specific population is greater than or equal to this random value, update the $X_{gbest}$;

Step6: Evolve the parent population according to formula (10), update the position and generate the offspring population;

Step7: Calculate the optimal fitness value after each update, update the temperature of the simulated annealing algorithm. Output the optimal particles after meeting the conditions of the number of iterations or convergence accuracy.

This article applies the SA-QUATRE algorithm introduced above to the coordinated design of power system stabilizers. For a multi-machine system, all the PSS parameter tuning and optimization processes are carried out simultaneously. The parameters to be tuned for each PSS are $T_1$ and $T_3$. Figure 4 shows the flow chart of optimizing PSS parameters by SA-QUATRE algorithm. Under the objective function and constraint conditions of Equation (6), the optimization steps are summarized as follows:

(1) First, import the basic power flow and dynamic data of the grid and configure the number and position of power system stabilizers according to the electromechanical oscillation characteristics of the power system to be analyzed. In the power system, installing power system stabilizers for each generator is unrealistic. So, the residue method is used to choose the installation location of PSS in this article [22];

(2) Set the parameters and operating mode of the power system stabilizer, linearize the system model and calculate the initial damping ratio in this mode through eigenvalues. If all are greater than $\xi_{\min}$, then end, otherwise continue;

(3) Initialize the parameters of the SA-QUATRE algorithm and simulated annealing, and at the same time, generate a set of initial solutions $X = \{X_1, X_2, \ldots, X_D\}^T$;

(4) Assign each solution $X_i$ to the variable in the PSS and calculate the damping coefficient of each electromechanical oscillation mode under this operating state;

(5) Evaluate the group according to the objective function based on characteristic value;

(6) Use the improved quasi-affine transformation evolutionary algorithm to continuously update and search, generate offspring populations, and update next-generation candidate solutions;

(7) The operation ends when reaching the maximum number of iterations, otherwise, it returns to the (4) step and enters the next cycle;

(8) Finally, obtain the global optimal parameter combination of PSS and the fitness value of the objective function. The specific process is as follows:

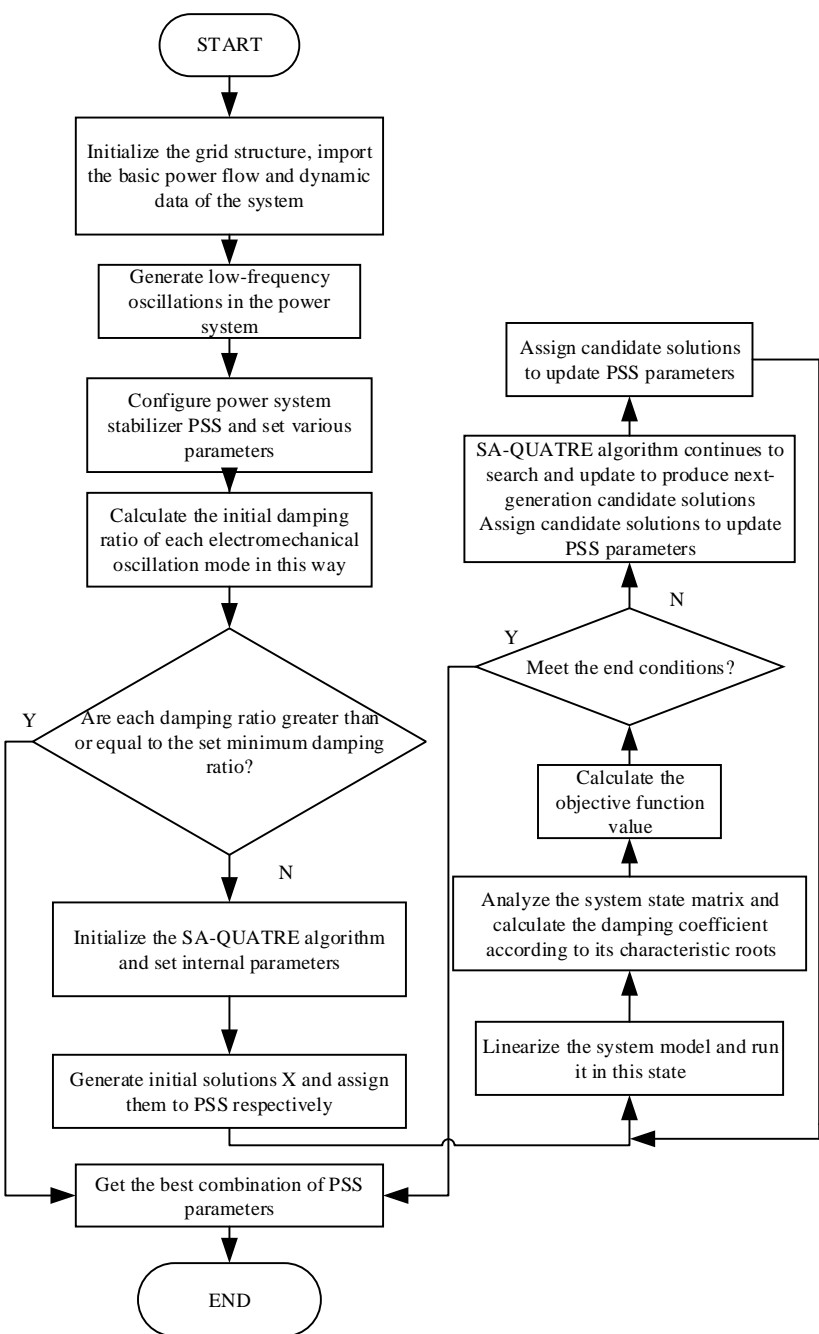

**Figure 4.** Flow chart of optimizing PSS parameters by SA-QUATRE algorithm.

## 4. Simulation Example

### 4.1. Algorithm Robustness Analysis

In order to preliminarily verify that the improved QUATRE algorithm in this paper has better convergence accuracy and global search ability, eight internationally-used benchmark functions were used for testing. The function expressions are shown in Table 1. The number of populations of different algorithms is 25, the maximum number of iterations is 1000, and the dimension of the function is 10. Figure 5 shows the convergence curve of the function. It can be seen that the improved QUATRE algorithm can approach the theoretical optimal value at the beginning of the iteration, which shows its fast convergence.

**Table 1.** Eight general benchmark functions.

| Functional Expression | Search Area | Optimum Value |
|:---:|:---:|:---:|
| $f_1 = \sum\limits_{i=1}^{D} x_i^2$ | $[-100,100]$ | 0 |
| $f_2 = x_i^2 + 10^6 \sum\limits_{i=2}^{D} x_i^2$ | $[-100,100]$ | $0^1$ |
| $f_3 = 10^6 x_i^2 + \sum\limits_{i=2}^{D} x_i^2$ | $[-100,100]$ | 0 |
| $f_4 = \sum\limits_{i=1}^{D} \left( x_i^2 - 10\cos(2\pi x_i) + 10 \right)$ | $[-100,100]$ | 0 |
| $f_5 = \sum\limits_{i=1}^{D-1} \left( 100\left(x_i^2 - x_{i+1}\right)^2 + (x_i - 1)^2 \right)$ | $[-100,100]$ | 0 |
| $f_6(x) = \frac{1}{4000}\sum\limits_{i=1}^{D} x_i^2 - \prod\limits_{i=1}^{D} \cos\left(\frac{x_i}{\sqrt{i}}\right) + 1$ | $[-100,100]$ | 0 |
| $f_7 = -20\exp\left(-0.2\sqrt{\frac{1}{D}\sum\limits_{i=1}^{D} x_i^2}\right) - \exp\left(\frac{1}{D}\sum\limits_{i=1}^{D}\cos(2\pi x_i)\right) + 20 + e$ | $[-100,100]$ | 0 |
| $f_8 = \left(\frac{1}{D-1}\sum\limits_{i=1}^{D-1}\left(\sqrt{x_i} + \sqrt{x_i}\sin^2\left(50x_i^{0.2}\right)\right)\right)^2$ | $[-100,100]$ | 0 |

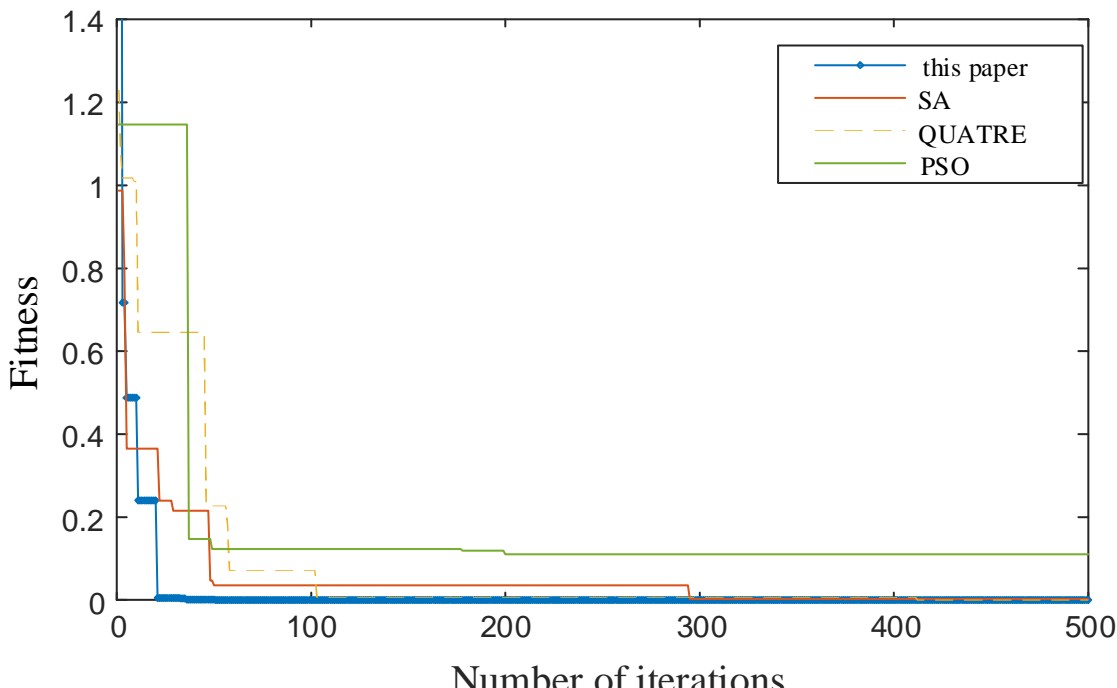

**Figure 5.** Iterative optimization of $f_1$ test function.

In order to avoid the chance of experiments, each function ran 50 times independently. The average value and error result of each algorithm are shown in Table 2. It is clear from the data in the table that the optimal fitness value of this method is the closest to the target value and its standard deviation is the smallest compared with the other three algorithms. In general, the algorithm in this paper has the ability of global optimization, and the overall performance of the algorithm has a certain superiority.

**Table 2.** Results of eight general benchmark functions.

| Function | SA-QUATRE | | PSO | | QUATRE | | SA | |
|---|---|---|---|---|---|---|---|---|
| | Mean Value | Standard Deviation | Mean Value | Standard Deviation | Mean Value | Standard Deviation | Mean Value | Standard Deviation |
| $f_1$ | 0 | 0 | 0.0012 | 0.0032 | $1.08 \times 10^{-26}$ | $3.10 \times 10^{-25}$ | $1.69 \times 10^4$ | 262.974 |
| $f_2$ | 0.2961 | 1.2047 | $3.82 \times 10^8$ | $8.52 \times 10^8$ | 0.3430 | 1.3370 | $2.90 \times 10^{19}$ | $1.53 \times 10^{19}$ |
| $f_3$ | $3.09 \times 10^{-7}$ | $3.05 \times 10^{-7}$ | 854.782 | 1483.98 | $3.403 \times 10^{-7}$ | $5.441 \times 10^{-7}$ | $6.51 \times 10^6$ | $3.80 \times 10^6$ |
| $f_4$ | 0.0484 | 0.0195 | $1.00 \times 10^6$ | $4.01 \times 10^5$ | 0.6340 | 0.5477 | $1.17 \times 10^7$ | $1.02 \times 10^6$ |
| $f_5$ | 2.7424 | 3.7638 | 16.073 | 22.983 | 3.7441 | 4.7638 | 1516.16 | 51.090 |
| $f_6$ | 0.1555 | 0.0886 | 2.8808 | 3.1924 | 0.2010 | 0.1773 | $2.881 \times 10^3$ | 74.165 |
| $f_7$ | 20.061 | 0.0777 | 20.323 | 0.0875 | 20.454 | 0.0920 | 20.556 | 0.0880 |
| $f_8$ | 0.7753 | 2.6725 | 48.426 | 22.262 | 0.8890 | 3.5521 | $1.35 \times 10^7$ | $3.41 \times 10^6$ |

*4.2. IEEE 4-Machine 2-Area System*

In order to verify the feasibility of the method proposed in this article in suppressing oscillations, the IEEE 4-machine 2-area system [23] is now used as an example, as shown in Figure 6. The generators $G_1$ and $G_2$ are installed in area 1. The generators $G_3$ and $G_4$ are installed in area 2, and a tie line connects the two areas. Because the two regions are connected by a contact line, the stability of the system itself is very weak. As long as there is a little disturbance outside the system, static instability easily occurs, and even leads to dynamic instability. Therefore, in practice, the system is always monitoring and protecting; it is sensitive and complex. After calculation and analysis, the PSS is configured in $G_2$ and $G_4$, and the dimension of the parameters to be optimized is six, which include the gain ($K_s \in [0.1, 50]$) and phase compensation ($T_1 \in [0.01, 1]$, $T_3 \in [0.01, 1]$) of each PSS respectively.

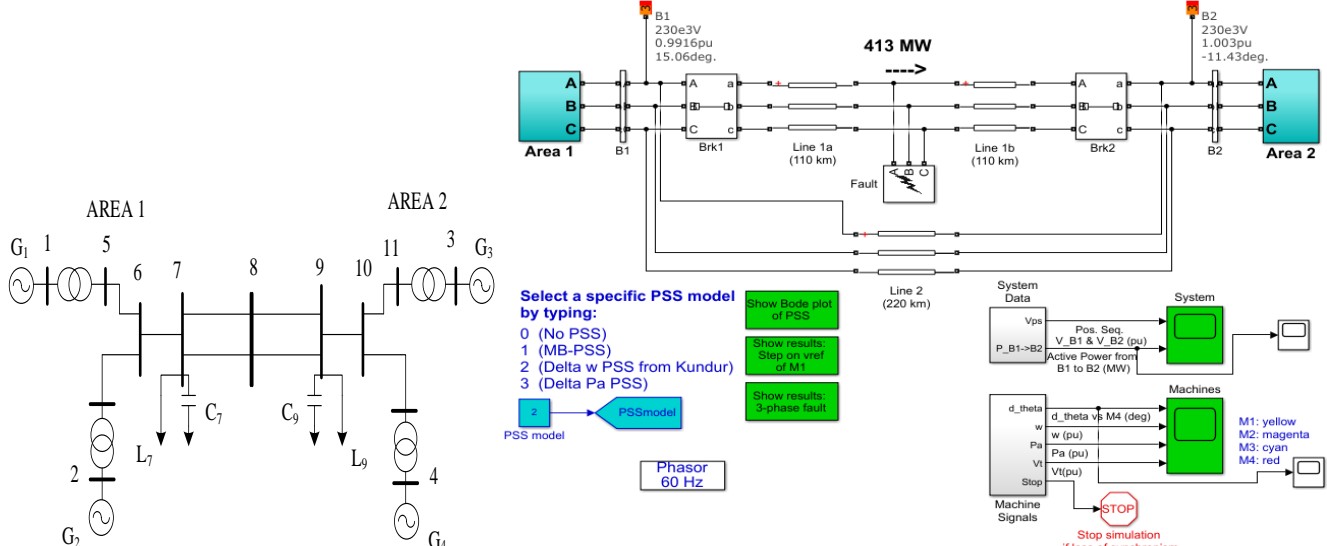

**Figure 6.** IEEE 4−machine 2−area system, structural diagram and Simulink model.

The effectiveness of the proposed method is analyzed in the following two operation states of the system.

4.2.1. Small Disturbance

At the excitation reference voltages on the generators $G_2$ and $G_4$, a square wave pulse disturbance signal with an amplitude of 0.05 appears at 1 s, and the duration is 0.1 s. In this disturbing operation mode, the electromechanical modes without power system stabilizers are shown in Table 3. It can be seen that the system has three oscillation modes, of which, the two modes with a frequency of about 1.1 Hz are regional modes, and the one with a frequency of 0.6409 Hz is the interval mode. The eigenvalues of mode 1 are in the right half

of the complex plane, showing negative damping, which is the most crucial control target in the coordinated design of multi-machine PSS.

**Table 3.** Electromechanical modes without PSS.

|  | **Mode 1** | **Mode 2** | **Mode 3** |
|---|---|---|---|
| characteristic root | $0.1029 \pm 4.0251i$ | $-0.6743 \pm 7.2483i$ | $-0.6804 \pm 7.0347i$ |
| $f$/Hz | 0.6409 | 1.1542 | 1.1202 |
| $\xi$ | $-0.0256$ | 0.0926 | 0.0927 |

The simulation time of the model is 15 s, and the objective function is iterated 1000 times with different algorithms. Finally, the global optimal PSS parameter combination is searched for as shown in Table 4. In this operating mode, the system tie-line oscillation power after each method is adjusted for PSS as shown in Figure 7. The *G*1 power angle signal (take *G*4 as a reference) is shown in Figure 8.

**Table 4.** Electromechanical modes without PSS.

| **Method** | $G_2$ | | | $G_4$ | | |
|---|---|---|---|---|---|---|
|  | $K_S$ | $T_1$ | $T_3$ | $K_S$ | $T_1$ | $T_3$ |
| PSO | 49.94 | 0.57 | 0.99 | 49.74 | 0.39 | 0.99 |
| SA | 46.05 | 0.81 | 0.96 | 42.26 | 1.00 | 0.58 |
| QUATRE | 42.67 | 0.65 | 0.66 | 38.60 | 0.75 | 0.39 |
| This paper | 49.66 | 0.96 | 0.693 | 49.146 | 0.516 | 0.98 |

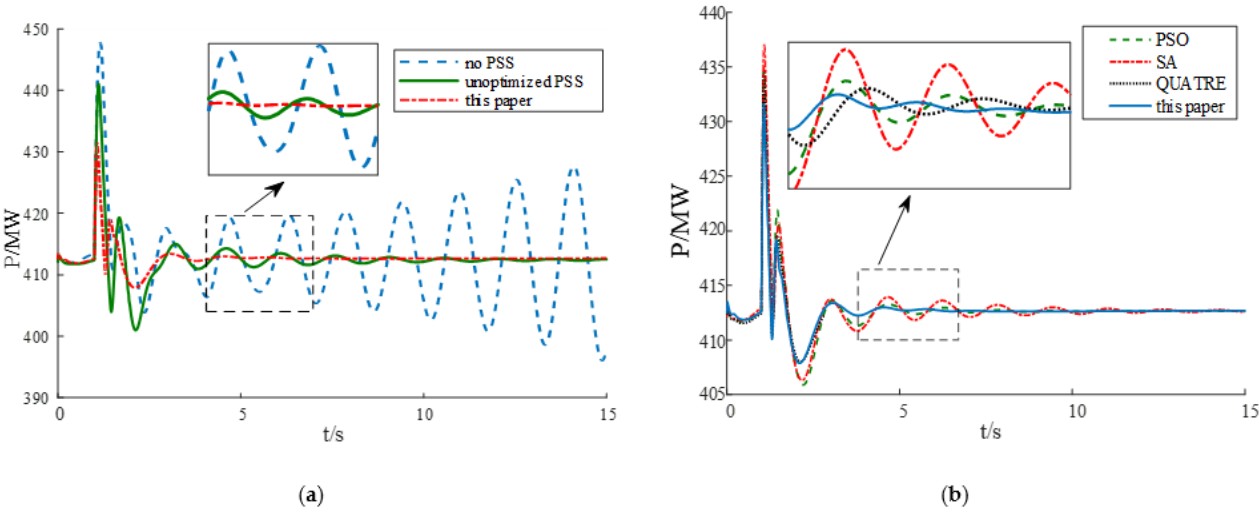

**(a)** **(b)**

**Figure 7.** Active power output of tie-line: (**a**) Comparison between the method in this paper and the unoptimized PSS; (**b**) Comparison of multiple algorithms to optimize PSS.

After a small disturbance is applied, the power on the tie-line between the two regions of the system will increase and oscillate. If it is not suppressed, the system will be seriously destabilized. Under the PSS parameters under the system default parameter configuration, that is, when the unoptimized PSS is applied, the power oscillation is suppressed to a greater extent than when the PSS is not installed, but the time required is about 13 s. The PSS, under the coordinated design of SA-QUATRE, can effectively calm down in the early stage of oscillation and reflect the rapidity of this method. From Figures 7 and 8, we can also see that, in comparison with other methods, before the oscillation is basically subsided, the overall amplitude of the PSS optimized by the method in this paper is more minor. In Figure 7a, you can see that under the traditional lead-lag compensators phase compensation setting, the power oscillation amplitude of the PSS appears as a local downdraft at 3~4 s,

indicating that the overshoot is too large and the system damping is relatively tiny. This phenomenon is effectively alleviated after optimization. In summary, the PSS parameters and suppression effect designed by the algorithm in this paper are more superior.

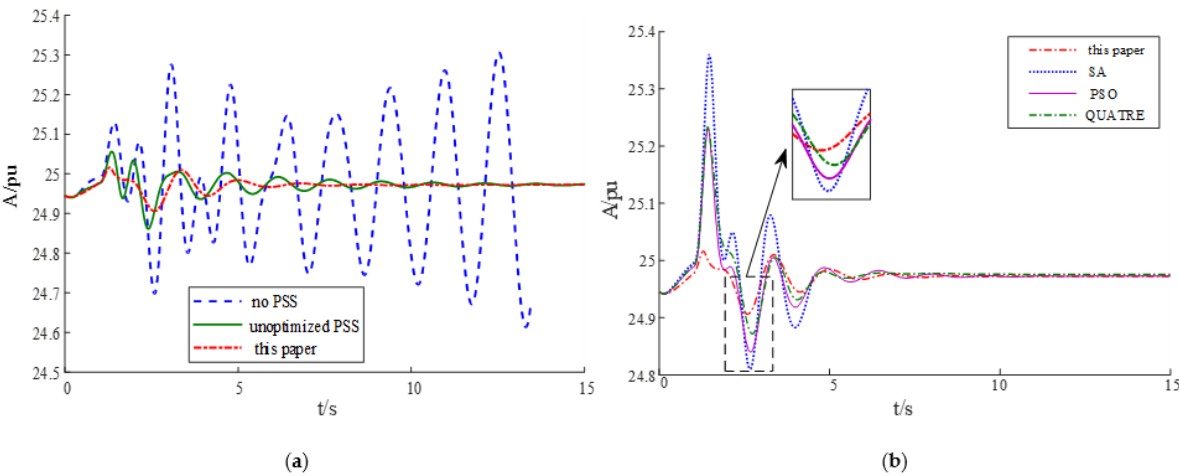

(a)

(b)

**Figure 8.** Power angle signal: (**a**) Comparison between the method in this paper and the unoptimized PSS; (**b**) Comparison of multiple algorithms to optimize PSS.

Use the above methods to coordinate the design of PSS parameters and compare them with the case where PSS is not installed. Since the suppression effect and the speed changes of generators *G*1~*G*4 are similar, only *G*1 speed changes are attached in this article, as shown in Figure 9. In the absence of PSS, the oscillation of the system cannot be attenuated to equilibrium, but it can be effectively suppressed with the stabilizer. The proposed method is better for the synthesis of the inhibition time and the overall amplitude. Table 5 shows the characteristic roots and damping of the system after the PSS function under the coordinated design of various methods. From the table, we can see that the installation of PSS can generally enhance the damping of each oscillation mode of the system, especially the dominant mode, that is, the negative damping mode mentioned in Table 2. It is further verified that the system after the PSS optimized by the method in this paper is installed under the disturbance of this method can have higher stability.

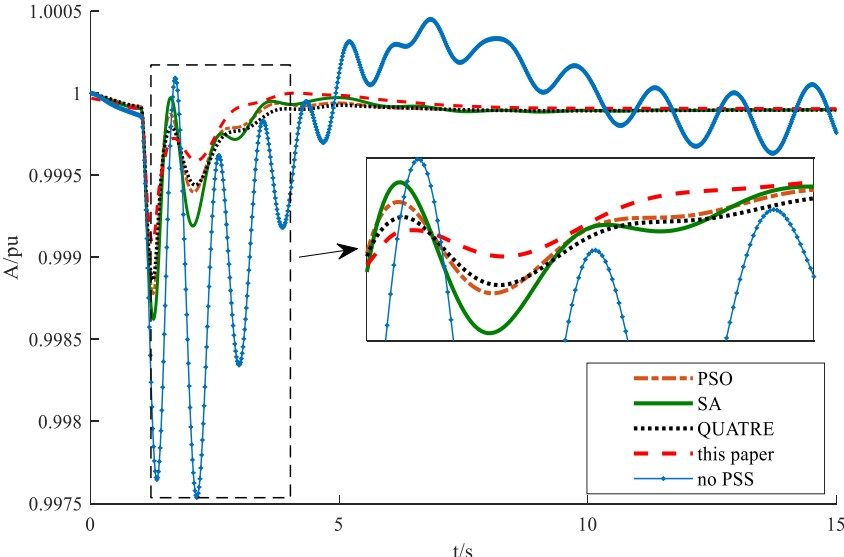

**Figure 9.** IEEE 4-machine 2-area system.

**Table 5.** Characteristic root and damping ratio of the system after designing PSS by different methods.

| Type | Characteristic Root | Damping Ratio | Minimum Damping Ratio |
|---|---|---|---|
| this paper | $-1.6173 \pm 5.2151i$ <br> $-1.3001 \pm 4.9332i$ <br> $-0.7476 \pm 3.7957i$ | 0.2962 <br> 0.2548 <br> 0.1933 | 0.1933 |
| QUATRE | $-1.6761 \pm 5.3423i$ <br> $-1.4534 \pm 4.5378i$ <br> $-0.6497 \pm 3.5789i$ | 0.2994 <br> 0.3050 <br> 0.1786 | 0.1786 |
| SA | $-1.8203 \pm 6.3656i$ <br> $-0.2659 \pm 3.9544i$ <br> $-0.5773 \pm 0.8160i$ | 0.2749 <br> 0.0671 <br> 0.1932 | 0.0671 |
| PSO | $-1.9048 \pm 5.6322i$ <br> $-1.8905 \pm 4.7963i$ <br> $-0.5381 \pm 3.9493i$ | 0.3204 <br> 0.3667 <br> 0.1350 | 0.1350 |

In order to see the system changes more intuitively, Figure 10 shows the eigenvalue distribution of the electromechanical mode when the system is configured with the PSS optimized by the method in this paper and when the PSS is not installed. It can be clearly seen from the figure that in the state of no PSS, the electromechanical modes are close to the imaginary axis and even reach the right half of the complex plane, indicating that the system has been unstable or has reached a critical stable state. After configuring the stabilizer set according to the method in this paper, the eigenvalues are all moved to the desired area, which enhances the damping ratio of each mode.

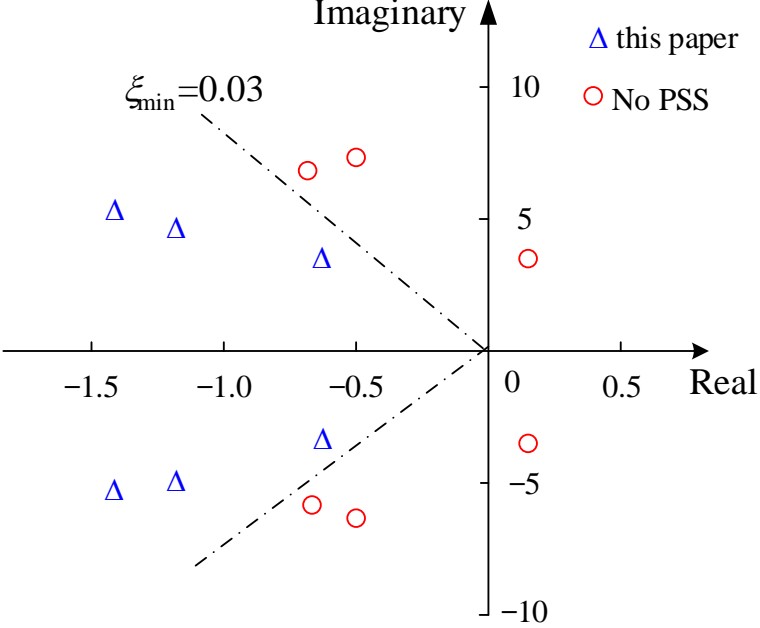

**Figure 10.** Distribution of eigenvalues.

### 4.2.2. Large Disturbance

This paper designs and verifies the transient performance of the power system stabilizer to further prove the applicability of the method proposed in this study. For the system shown in Figure 6, a large disturbance is set, in which occurs a three-phase short-circuit at 1 s that stops at 1/6 s. To coordinate the design of the PSS, the method described in this paper and the traditional phase compensation method were used, respectively, the comparison was made with the controller not the installed scenario. Figures 11 and 12 depict the change in active power on the tie-line and the rotation speed of generator *G*1 in

this mode of operation. Figure 13 depicts the *G2* power angle signal change using the *G4* power angle as a reference.

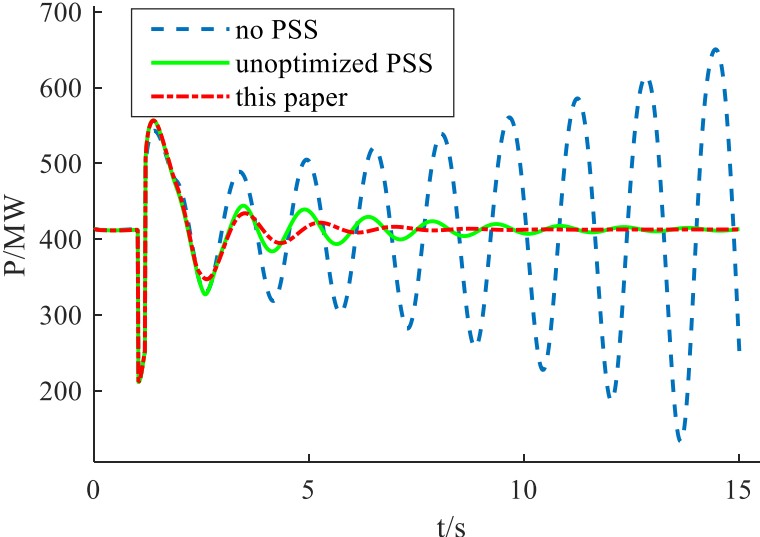

**Figure 11.** Active power output of tie-line.

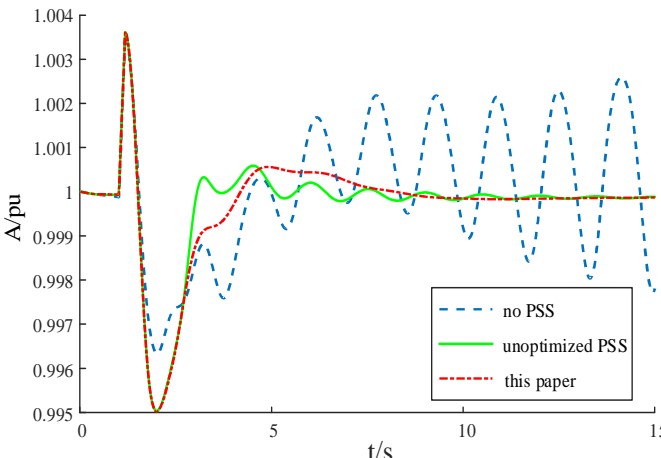

**Figure 12.** Speed change.

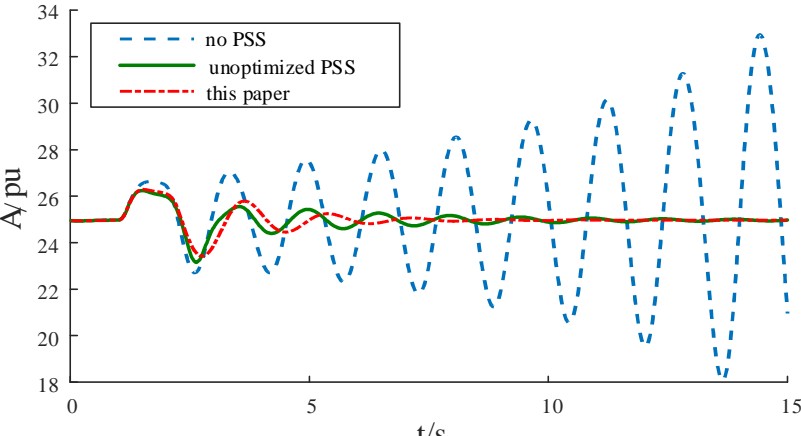

**Figure 13.** Power angle signal.

In the case of this large disturbance, the system has an amplitude-enhancing oscillation, and the PSS adjusted by the traditional phase compensation method is adopted. Although it can attenuate the oscillation to balance within a certain period of time, it is slightly inferior in speed and amplitude compared to the coordinated design method proposed in this article. The power system stabilizer optimized in this paper can effectively suppress it quickly, and the overall amplitude of the oscillation is smaller in the early stage of oscillation. Table 6 shows the various electromechanical modes of the system with the power system stabilizer designed by this method and without PSS. The table shows that in this disturbance operation state, a negative damping mode will be generated when the stabilizer is not configured, which threatens the system's stable operation. After inputting the PSS tuned by the method proposed in this paper, not only is the negative damping of the dominant oscillation mode increased to the positive damping, but the damping of the other two modes is also enhanced at the same time.

**Table 6.** System characteristic root and damping without and with PSS installed.

| Mode | Not Installed PSS Characteristic Root | Damping Ratio | Method of This Article Characteristic Root | Damping Ratio |
|---|---|---|---|---|
| 1 | $0.1051 \pm 4.0277i$ | $-0.0261$ | $-0.6497 \pm 3.5789i$ | $0.1786$ |
| 2 | $-0.6683 \pm 7.2631i$ | $0.0916$ | $-1.6173 \pm 5.2152i$ | $0.2962$ |
| 3 | $-0.6756 \pm 7.0499i$ | $0.0954$ | $-1.3001 \pm 4.9333i$ | $0.2548$ |

The main objective of this paper is to maximize the damping ratio by moving the eigenvalues of the system to the left of the imaginary axis. From the characteristic distribution of electromechanical mode in Figure 14, it can be seen clearly and intuitively that the coordinated design of PSS using the method in this paper can also achieve the goal and effectively improve the system's stability in a large disturbance state.

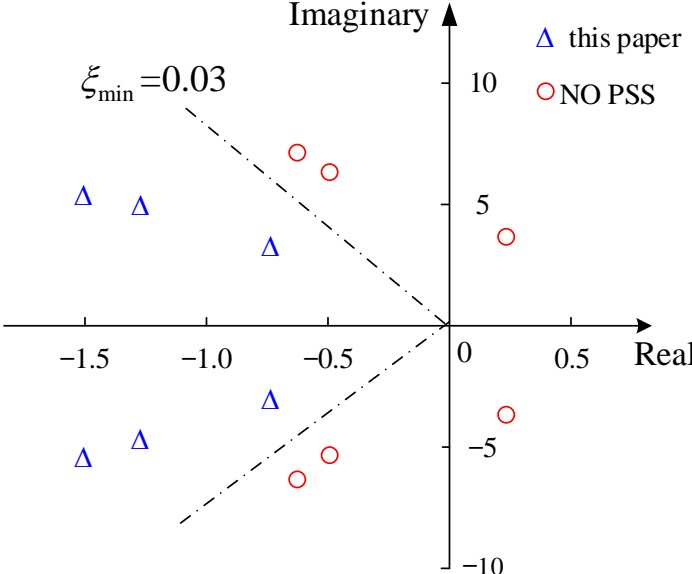

**Figure 14.** Distribution of eigenvalues.

## 5. Conclusions

This paper proposes a multi-machine power system stabilizer parameter coordinated optimization design based on the SA-QUATRE algorithm to suppress power system low-frequency oscillation and enhance system stability. For the first time, the new algorithm is applied to power system problems, and the following conclusions are reached through experiments:

(1) In the case of system oscillation instability, the installation of the power system stabilizer on the generator can effectively calm the oscillation, so that the system can be restored to normal operation in a specified time. However, the effect of PSS presented under different design methods is also different. Simulations show that using the method in this paper to design the parameters of PSS can convert the negative damping electromechanical mode that threatens the stability of the system to positive damping, and at the same time enhance the damping ratio of other oscillation modes, effectively improving the stability and robustness of the system, regardless of the operation mode of small or large disturbance;

(2) For multiple power system stabilizers in the system, the traditional phase compensation method usually carries out parameter settings sequentially, which will inevitably cause the interaction effect between each stabilizer. This paper uses the performance target based on eigenvalues instead of the traditional phase compensation method to optimize the PSS parameters to achieve more effective suppression of the low-frequency oscillation of the power system;

(3) The SA-QUATRE algorithm proposed in this paper can jump out of the local optimum and has better convergence and accuracy. This algorithm not only overcomes the defect of slow convergence of the particle cluster optimization algorithm, it also has stronger cooperation, reduces time complexity, and accepts inferior solutions with a certain probability to avoid falling into local minimum in the process of searching;

(4) The SA-QUATRE algorithm has good applicability. The applicability of this method to the coordinated design of the damping controller of the flexible AC transmission system is still under study, and it is also worthy of further popularization and application to other fields;

(5) In the future, PSS can be used to suppress the low-frequency oscillation caused by the addition of renewable energy to the traditional power grid and analyze the adaptability of PSS in new power systems.

**Author Contributions:** Formal analysis, C.Z.; Funding acquisition, J.H.; Investigation, J.L. and Y.K.; Methodology, J.H. and S.W.; Software, J.L.; Supervision, J.H. and C.Z.; Writing—original draft, Y.K.; Writing—review & editing, S.W. All authors have read and agreed to the published version of the manuscript.

**Funding:** This research was funded by National Natural Science Foundation of China, grant number 51977039, grant number 61872095, grant number 61571139, grant number 61872128, in part by Fujian Institute of Technology Research Startup Fund Project, grant number GY-Z18060, in part by Fujian Science Fund for Distinguished Young Scholars, grant number 2020J06043.

**Institutional Review Board Statement:** Not applicable.

**Informed Consent Statement:** Not applicable.

**Data Availability Statement:** The data used in this study are available on request from the corresponding author.

**Conflicts of Interest:** The authors declare no conflict of interest.

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
