# Peer review of "Application of Improved Quasi-Affine Transformation Evolutionary Algorithm in Power System Stabilizer Optimization"

_electronics, doi:10.3390/electronics11172785_

Round 1
Reviewer 1 Report
The topic is interesting. But more works need on this paper for further consideration.
1. The title is appropriate.
2. The paper needs to be proof-read, completely.
3. The abstract should focus on the paper novelties and important findings. In current form it is general and should be revised significantly. Adding some quantitative results is recommended. Please revise the abstract section completely.
4. A Section describing all nomenclature, parameters and variables should be added to the paper. The relevant dimensions should be added to nomenclature section.
5. Literature review should be completed and enriched. Bulk referencing should be avoided. The Introduction should be sensibly improved by presenting the application context, the open problems, the relevant works, and the paper contributions. Adding some new referencing on the paper topic, especially for 2020-2022 years is strongly recommended.
6. An important question: why all of the references are limited to a specific country? This manner should be corrected at all.
7. The authors encouraged to include some updated references regarding the comparative study on evolutionary algorithms for optimizing BESS considering the impacts of wind power plants, and the positive effects of SVR on voltage stability improvement of wind farms by ESS.
8. The contributions should be bolded and highlighted.
9. The paper structure should be added to the end of introductions
10. Please merge sections 2, and 3.
11. All Figs. are not clear. Should be improved completely.
12. Please merge sections 4, and 5.
13. The authors should justify the procedure for selecting the setting parameters.
14. All needed formulation should be presented in section 2.
15. There are many equations and formulations with a little description in some cases. Please comment on this subject.
16. There are many equations without referencing. All of these types of equations should be cited appropriately.
17. What are the main steps of the proposed strategy?
18. The used software is not clear.
19. What is the reference for simulation data? based on some experimental analysis? Please clear this subject. Many of the dimensions are neglected.
20. The impact of data uncertainty on the obtained solutions should be accurately discussed, in this context the integration of some kind of robustness measures as one of the most promising research direction. The authors are invited to clarify how the proposed method manages data uncertainty and how the robustness of the obtained solution is assessed.
21. More details of the algorithm complexity and applied software are required.
22. Discussion and conclusion sections are weak and not acceptable. Should be extended.
23. Add some future scope of the work with the conclusion section.
Reviewer 2 Report
The manuscript entitled “Application of improved quasi-affine transformation evolutionary algorithm in power system stabilizer optimization” has been investigated in detail. The topic addressed in the manuscript is potentially interesting and the manuscript contains some practical meanings, however, there are some issues which should be addressed by the authors:
1) In the first place, I would encourage the authors to extend the abstract more with the key results. As it is, the abstract is a little thin and does not quite convey the interesting results that follow in the main paper. The "Abstract" section can be made much more impressive by highlighting your contributions. The contribution of the study should be explained simply and clearly.
2) The readability and presentation of the study should be further improved. The paper suffers from language problems.
3) The “Introduction” section needs a major revision in terms of providing more accurate and informative literature review and the pros and cons of the available approaches and how the proposed method is different comparatively. Also, the motivation and contribution should be stated more clearly.
4) The importance of the design carried out in this manuscript can be explained better than other important studies published in this field. I recommend the authors to review other recently developed works.
5) What makes the proposed method suitable for this unique task? What new development to the proposed method have the authors added (compared to the existing approaches)? These points should be clarified.
6) Equation 11 should be checked.
7) “Discussion” section should be added in a more highlighting, argumentative way. The author should analysis the reason why the tested results is achieved.
8) The authors should clearly emphasize the contribution of the study. Please note that the up-to-date of references will contribute to the up-to-date of your manuscript. The study named "Model predictive control of three-axis gimbal system mounted on UAV for real-time target tracking under external disturbances; Metaheuristic optimization-based path planning and tracking of quadcopter for payload hold-release mission" recently published in this journal - can be used to explain the proposed method and optimization process in the study or to indicate the contribution in the “Introduction” section.
9) How to set the parameters of proposed method for better performance?
10) The complexity of the proposed model and the model parameter uncertainty are not enough mentioned.
11) It will be helpful to the readers if some discussions about insight of the main results are added as Remarks.
This study may be proposed for publication if it is addressed in the specified problems.
Round 2
Reviewer 2 Report
My comments have been thoroughly addressed. It is acceptable in the present form.
Author Response
Thank you very much for your careful review.